# Evaluation of Community Pharmacists’ Competences in Identifying and Resolve Drug-Related Problems in a Pediatric Prescription Using the Simulated Patient Method

**DOI:** 10.3390/pharmacy11010006

**Published:** 2022-12-30

**Authors:** Riham M. Hamadouk, Fatimah M. Mohammed, Esra D. Albashair, Bashir A. Yousef

**Affiliations:** 1Department of Clinical Pharmacy, Faculty of Pharmacy, University of Khartoum, Khartoum 11111, Sudan; 2Department of Pharmacology, Faculty of Pharmacy, University of Khartoum, Khartoum 11111, Sudan

**Keywords:** drug-related problems, community pharmacists, pediatrics, simulated patient

## Abstract

**Background:** Drug-related problems (DRPs) are a global issue that impacts the efficacy and safety of the therapy, and pediatric patients are considered to be more vulnerable to DRPs, thus requiring more attention. Community pharmacists (CPs) are in a position that allow them to identify and alleviate these DRPs. **Objectives:** This study evaluated the ability of CPs in identifying and resolving DRPs in a pediatric prescription. **Methods:** A cross-sectional study was carried out in 235 community pharmacies to evaluate the ability of CPs working in the Khartoum locality to identify DRPs in a pediatric prescription and how they intervene to resolve these problems. Fifth-final year B. Pharm. Students were selected and trained to act as simulated patients (SPs) for this study. The visits were performed by using a simulated prescription that contains three different types of DRPs. The information obtained from the visits was documented immediately by the SPs after leaving the pharmacy in a data collection form. **Results:** All planned SPs visits were completed. Of the 235 community pharmacies, only 50 (21.3%) CPs were able to identify at least one of the DRPs. The most common type of DRP identified was the wrong duration of the treatment 19%, followed by the wrong dose 4%. The interventions made by CPs to mitigate the identified DRPs included recalculation and correction of the dose according to weight, which was made by 10 CPs, and correction of the duration, which was done by 45 CPs. None of the CPs who identified the presented DRPs communicated with the physician or referred the SP to the prescriber. The average dispensing time of the CPs was 68.18 ± 36.1 s. **Conclusions:** The majority of the CPs in the Khartoum locality were unable to identify DRPs in a pediatric prescription. Correction of the dose and duration of treatment were from the attempts of CPs to resolve DRPs. However, no collaboration was observed between CPs and physicians. In general, the practice of CPs in Khartoum locality in this area requires substantial improvement.

## 1. Introduction

Community pharmacists (CPs) are an indispensable asset to healthcare. Their patient care role in the healthcare delivery system cannot be denied [1]. They are highly educated with professional skills and training that allow the provision of optimum health care services. Their unique position in the communities eases their accessibility to patients and other members of the community [2].

Around the world, their role has been extended to encompass chronic disease management, weight management, immunization services, and providing advice to other healthcare members [3]. Despite that, dispensing is considered one of the basic processes performed by community pharmacists. During dispensing, their role includes providing an independent assessment of prescriptions away from the prescribing process before treatment starts. Throughout their assessment and interpretation of the prescription, they check the appropriateness of the treatment to the diagnosis, the suitability of the dosage for the patient, detect the drug interactions and contraindications, and resolve any drug-related problems before the medicines are supplied to the patient [4,5].

Drug-related problems (DRPs) are, according to The Pharmaceutical Care Network Europe (PCNE), “an event or circumstance involving medication treatment that actually or potentially interferes with the patients’ desired health outcomes”. Several domains have been described by PCNE to cause DRP; some of these are drug selection, dosage form, dose selection, treatment duration, and others [6]. DRPs can be an incorrect dosage (overdose or sub-therapeutic dose), taking a drug for no reasonable medical indication, inappropriate selection of a drug, improper treatment duration, drug interactions, adverse drug reactions, and patient-related reasons like psychological and sociological issues [7]. DRPs are a worldwide problem that affects the safe and effective provision of medication; one of the most affected populations is the pediatric population [8].

It is known that childhood is a period of rapid growth associated with various changes in the development of the internal organs and in the activity of the hormones and enzymes that handle drugs [9]. As the pharmacokinetics profile of pediatrics varies during the process of their development, drug absorption, distribution, and elimination can be remarkably affected [10]. Thus, the drug dose should be calculated to be adjusted according to the child’s body weight, however, the maturity of the child should be taken into account as well [11].

Children are considered vulnerable to DRPs, and any error that might be tolerated in adults can result in considerable consequences on the child’s health [12]. One study has shown that almost 5691 pediatric patients who visited the emergency room in 2004 were a result of medication errors, and the predominant error was prescribing incorrect doses by the physician [13]. In another study, prescribing error was detected in 16% of pediatric outpatient prescriptions, and improper dosing was among the most detected medication errors [14].

In Sudan, there is only one study that addressed DRPs. The study was in the adult patient population at a tertiary hospital, and it revealed that the most common DRPs detected were incomplete drug therapy, inappropriate drug use, and high drug dose [15]. Therefore, because the pediatric population is more prone to DRPs, and community pharmacists are considered important healthcare members who help in preventing the incidence of DRPs, the current study aims to evaluate the community pharmacists’ competence in detecting and resolving DRPs in pediatric prescription.

## 2. Materials and Methods

### 2.1. Study Design

A cross-sectional design was used in this study agreeing with the STROBE statement that must be included in reports of cross-sectional studies [16] to evaluate the ability of community pharmacists working in Khartoum locality in identifying DRPs in a pediatric prescription, and how they intervene to resolve these problems. The simulated patient approach was employed in this study because it is a well-validated and globally established method for evaluating community pharmacists’ professional performance. This method uses trained individuals to act as test buyers to present a specific scenario resembling a real-world situation through which the quality of the services can be assessed [17]. The main rationale behind using this approach is to minimize the Hawthorne effect (people change their behavior upon awareness of being tested) [18].

### 2.2. Scenario

One scenario was developed by the researchers in consultation with two members of the university of Khartoum: a pharmacy practice educationalist and a pediatric specialist. The paper prescriptions used in this simulation were obtained from a well-known hospital in the Khartoum locality. The condition and the medications in the scenario were selected depending on the most common conditions and medications that the CPs encounter in their daily practice. The most common conditions and medications were identified from data collected previously through questionnaires for another study.

There is no global standard for a prescription format and they vary from country to another, but there are legal requirements for a valid prescription, and there is a global agreement that the age, gender, weight, and diagnosis of the patient should be stated in the prescription, especially for children [19]. Thus, a prescription was designed for a pediatric patient including information about the patient’s age, weight, diagnosis, and all necessary information such as the name and the telephone number of the prescriber (one of the researchers), which are required in a standard prescription, as seen in Figure 1. In addition to that, the information was written in clear handwriting. It covers three aspects of DRPs (wrong dose, wrong duration, and wrong indication). Then, the content of the prescription was validated by an expert physician and two pharmacists. In the scenario, the SP enters the pharmacy and provides the prescription to the pharmacist. The prescription for 3 year old boy diagnosed with otitis media, contained co-amoxiclav 400/57 mg (6 mL twice daily for 3 days) and metronidazole suspension 200 mg (3 mL three times per day for 5 days). If the pharmacist asks the SP about any other patient conditions that indicate the use of metronidazole, no helpful information will be provided. The scenario is demonstrated in Table 1.

### 2.3. Sample Size and Sampling

The Sudanese General Directorate of Pharmacy list, which includes all registered community pharmacies in the Khartoum locality, was used to select the community pharmacies enrolled in this study. The sample size was calculated, according to the equation: *n* = N/1 + N(e)^2^, where (e) an error margin of 0.05, the formula is based on a degree of variability of *p* = 0.5, and a 95% confidence interval. The number of registered community pharmacies in the Khartoum locality represents the study population (N), which is equal to 568. Using the equation, the necessary sample size required for this study (n) was 235. By applying a simple random sampling technique, 235 community pharmacies were selected.

### 2.4. Assessment Form

A data collection form was developed by the researchers to document the required information for the study. It was validated by three academic members from the University of Khartoum who had excellent experience in pharmacy practice. Then, the collection form was piloted in twenty-five community pharmacies. The form consists of three sections; the first one assesses the identification of DRPs which includes closed-end questions (yes/no). Section 2 later assesses the response of the community pharmacist upon DRPs identification. Open-ended questions were applied in this section. Finally, the form ends with the amount of time spent during the dispensing process (Table 2).

### 2.5. Data Collection

In this study, the simulated patients (SPs) were final-year students from the Faculty of Pharmacy University of Khartoum. Two males and three females were available for participation in this study as SPs. Before the data collection, the principles of the methodology were well explained to the SPs. They were also trained to act as patients and introduced to a real-life scenario. After the simulated patients were familiarized with the scenario and the contents of the assessment form, several rehearsals were conducted to ensure that they could identify the pharmacist practice in the particular situations. After that, a pilot study was carried out to ensure the readiness of the SPs in performing the method, and to confirm the functionality of the data collection form. A total of 25 community pharmacies were visited (five visits for each SP), and all the visited pharmacies were not part of the study sample. Prior to the visits, the community pharmacies were randomly distributed across the five SPs. The test visits were conducted between September 2021 and December 2021. At the time of the study, the disease control measures that used to control COVID-19 pandemic were difficult to implement in Sudan due to economic, social, and political challenges, and people in the Khartoum locality were living their normal daily lives. The pharmacies were approached only after ensuring that they were empty. Each pharmacy was visited by a SP and an observer (one of the researchers), whose main role was to measure the time spent by CP engaging with the SP using a stopwatch. Each SP made 47 visits to complete 235 pharmacy visits. In each visit, to ensure that the prescription was assessed by a pharmacist, the SP requested the pharmacist first before providing the prescription. The data collection form was completed immediately after leaving the pharmacy by the SP to minimize the recall bias and the time of dispensing recorded by the observer.

### 2.6. Data Management and Analysis

For the analysis of data, Statistical Package for Social Sciences software, version 23.0 (IBM SPSS Inc., Chicago, IL, USA) was used. Initially, all information was collected via a data master sheet and then coded into variables. Descriptive statistics involving frequency tables were used to present the results. The relationship between variables was tested using an appropriate statistical method (the Chi-square test) with a *p*-value of less than 0.05 being considered statistically significant.

## 3. Results

### 3.1. Socio-Demographic of the Community Pharmacists

A total of 235 pharmacy visits were accomplished by the SPs with a prescription containing three different types of DRPs, resulting in a completion rate of 100%. By using the prescription, the SPs tested the competence of the community pharmacists in identifying the DRPs and observed their resolution accordingly. Of the 235 community pharmacists, 72 (30.6%) were males and 163 (69.4%) were females. These CPs held a Bachelor degree as a minimum qualification to practice their job. Socio-demographic characteristics of the CP in Table 3.

### 3.2. Identification of DRPs by the Community Pharmacists

Of the 235 community pharmacists, 50 (21.3%) were able to identify at least one of the DRPs presented in the prescription, while 185 (78.7%) failed to identify any of them. The most common type of DRP identified was the wrong duration of the treatment, followed by the wrong dose. There is no significant association between DRPs identification by CPs and their gender, years of experience, and daily working hours in the pharmacy. The identification of DRPs is listed in Table 4.

### 3.3. Action Taken by the Pharmacists in Managing DRP

In resolving the DRPs presented to the community pharmacists, referring the patient to the prescriber was not done by any community pharmacist. However, appropriate actions were also sometimes taken; six (100%) refused to dispense the unnecessary medication with the wrong indication, and ten (100%) recalculated the dose according to weight and corrected the wrong dose. Results are shown in Table 5. Regarding the time spent by the community pharmacist during dispensing, the time was less than three minutes, and the average time was 68.18 ± 36.1 s.

## 4. Discussion

Preventing harm from medicines is one of the components that is required to achieve the mission of pharmacy practice in improving the health of patients, this harm may arise from DRPs causing ineffective therapy and may lead to drug-related morbidity and mortality [20,21]. Community pharmacists play a crucial role in preventing this harm because their responsibility includes ensuring that the required therapeutic target is achieved, by assuring the prescribed medicines with the appropriate dosages and dosage forms, providing clear instructions about the use of the drugs, decreasing the unnecessary treatments, and preventing any drug–drug and drug–food interactions [22,23].

Many studies focused on assessing hospital and clinical pharmacists’ ability in detecting and correcting DRPs in pediatric wards [24,25,26,27], and others showed the impact of clinical pharmacists in reducing DRPs and improving patient outcomes [28,29,30]. Unfortunately, only a few studies have previously assessed community pharmacists’ ability to detect and resolve DRPs. Therefore, it is difficult to compare the performance of participants in this study with similar previous estimates.

In the current study, the findings revealed that 78.7% of the community pharmacists under study were unable to identify any DRPs presented in the simulated prescriptions and they did dispense inappropriate medications. In our study, 96% of the CPs failed to identify the wrong dose, and 97% and 81% of them did not identify wrong indication and wrong duration, respectively.

The failure rate of identification is high comparing it with the simulated patient study from the United Arab Emirates conducted in 2020 that assessed the ability of community pharmacists to identify DRPs in prescriptions, where 56% of CPs also failed to identify DRPs. This study used prescriptions containing different five DRPs; among them were wrong drug dose, wrong duration of treatment, and wrong indication. A wrong dose was not identified by 50% of the CPs similarly for wrong duration of treatment, while 20% of the CPs failed to identify the wrong indication [31]. In another comparable study conducted in 2019 from the United States of America, the ability of community pharmacists was tested by using survey including three common pediatric prescriptions. These prescriptions focused on the inappropriate drug doses, and the study has concluded that community pharmacists did not consistently identify wrong drug doses in pediatric prescriptions [13].

Wrong medication dose as a high dose may lead to harmful effects, and if the dose is inadequate, the therapeutic target will not be achieved. Regarding pediatric dose, it is known that the majority of drug dosage depend on the child’s weight. A study conducted in France in 2018 found that several DRPs in the pediatric population were due to the absence of the weight of the child on the prescription [32]. Moreover, in another study, 68% of the pharmacists strongly agreed and 31% of the pharmacists agreed with the statement that when checking a pediatric dose, knowing the weight of a pediatric patient is helpful [33].

In this study, the weight of the child was written in the prescription, despite that only 10 (4%) CPs identified the wrong dose and recalculated the dose based on the weight and age of the child. This result was low compared to the study conducted in Jordan which assessed the knowledge of community pharmacists about appropriate dosing of antibiotics among pediatrics, where about 29% of CPs determined the correct dose of Co-amoxiclav in otitis media [34]. Furthermore, a higher result was observed in another study that aimed to measure the awareness and knowledge of Palestinian Pharmacists about pediatric doses using a questionnaire, where (51.7%) of the pharmacists gave the correct answer for the dose of Co-amoxiclav [35].

The wrong or inadequate duration of antibiotics is considered inappropriate antibiotic use. Inappropriate antibiotic use is the direct cause of antibiotic resistance, a nationwide issue that enhances the strength of infectious diseases resulting in increasing the hospitalization period, treatment costs, and mortality rate [36]. In this case, the diagnosis was otitis media and the recommended duration of Co-amoxiclav is 5 to 10 days. Furthermore, a previous report demonstrated that in the treatment of acute otitis media 5 days of antibiotics is as effective as 10 days in all children above 2 years [37]. In the present study 190 (81%) of the CPs failed to detect the wrong duration of the antibiotic, and the 45 (19%) CPs who detected this DRPs recommended continuing the medication for 5 days.

Antibiotic resistance is the utmost danger that threatens public health, especially children. According to World Health Organization (WHO) data, there are 700,000 deaths across all ages due to infections caused by multidrug-resistant bacteria. One of the causes of antibiotic resistance is the use of antibiotics for wrong diagnoses and indications [38]. In this study, metronidazole suspension was prescribed for a child diagnosed with otitis media. There is no rational reason for prescribing it in this case, so it is considered an inappropriate indication for diagnosis. Inappropriate indication for diagnosis is when a drug is prescribed to a patient that is not intended to treat the symptoms of the diagnosed disease and is not consistent with the drug indication [39].

Only 3% of the CPs identified the inappropriate indication of metronidazole and all refused to dispense the medication. However, 97% of the CPs did dispense the drug, putting the pediatric patient at risk of developing the side effects of the drug which include gastrointestinal disturbances, oral mucositis, anorexia, thrombocytopenia, rash, and jaundice [40].

Upon discovery of an inappropriate prescription that may contain an improper dose, unnecessary treatment, or interaction, the pharmacist should contact the prescriber to resolve the drug-related problem [41]. According to the literature, if any DRP is identified by community pharmacists, it is important to refer patients back to the prescriber [42]. Community pharmacists’ collaboration with the prescriber is very important as they may lack much patient information, especially about patient medical history and current medical problems, which may help them in identifying medication-related issues. What was noticeable in this study was that none of the community pharmacists referred the patient back to the physician or tried to contact the prescriber. On the other hand, in the study by Palaian et al., 38% of CPs referred simulated patients to a physician, of the thirty-eight 70% in response to prescriptions with the wrong indication, 50% in response to prescriptions with the wrong dose, and 50% in response to prescriptions with wrong duration [31]. The lack of communication in our study may be because pharmacists feel that prescribers do not like them to intervene in their decisions, and some doctors do not accept the pharmacist advice. Another reason might be pharmacists may miss the suitable way through which they can professionally approach prescribers without judging their knowledge. To establish a professional pharmacist–physician collaboration, the pharmacist’s responsibilities and duties should be clearly defined by The Sudanese General Directorate of Pharmacy and orient the health care professionals about the pharmacist roles. Further, if the shape of the collaboration determined at a level of legislation, this might facilitate the interprofessional collaboration.

Effective counseling required time. According to the WHO, pharmacists need to spend at least three minutes with every patient to provide the necessary orientation [43]. In this study, all of the CPs took less than three minutes to dispense the medication, which is very short comparing it to the standard counseling time of the WHO. Further, the time spent by pharmacists during counseling has been established in the literature to be substantially longer than this [44]. Similar observations of a short time spent by CPs during dispensing were noted in Qatar [45], United Arab Emirates [30], and Cyprus [43].

This study has some limitations. The study is a cross-sectional study, and only community pharmacies in the Khartoum locality were visited. Thus, the findings might not be able to be generalized to the entire country’s practice. In addition, as audio recording which helps in minimizing the recall bias was not used in this study, there is a possibility that some information might not be documented while completing the checklist. In addition to that, the current study did not assess the practice of community pharmacists regarding other types of DRPs like drug interactions and adverse drug reactions. Certain factors that might affect the pharmacist’s performance, such as their participation in post-graduations and training courses and previous experience in hospital and clinical settings, were not covered in this study. Further, the findings have reflected a specific scenario because only one simulated prescription with specific medications were used. Hence, it may be difficult to extrapolate the results of this study to the entire pharmacy practice of community pharmacists in Sudan.

The findings of this study recommend the need for improving community pharmacy practice standards. Continuing education is necessary for community pharmacists to enrich their knowledge and improve their skills. Thus mandatory educational programs should be endorsed by stakeholders. Such training should include a look at the most common pediatric prescriptions, focusing on confirming the accuracy of weight-based dosing, as well as ways for detecting and preventing DRPs in pediatric prescriptions. The findings also suggest further in-depth research should attempt to quantify the ability of community pharmacists in identifying and managing DRPs in pediatric prescriptions and other areas of their practice.

## 5. Conclusions

Although the pharmacist is the last safety guard of the health care system, this study has shown that the majority of the community pharmacists in Khartoum locality were unable to identify DRPs in a pediatric prescription. Correction of the dose and duration of treatment were from the attempts of CPs to resolve DRPs. However, no collaboration was observed between CPs and physicians to resolve the identified DRPs, as none of the pharmacists tried to contact the prescriber or referred the patient. The findings also revealed that CPs spent minimal time when counseling the patients. In general, the ability of community pharmacists to identify DRPs in a pediatric prescription in Khartoum locality was below expectation.

## Figures and Tables

**Figure 1 pharmacy-11-00006-f001:**
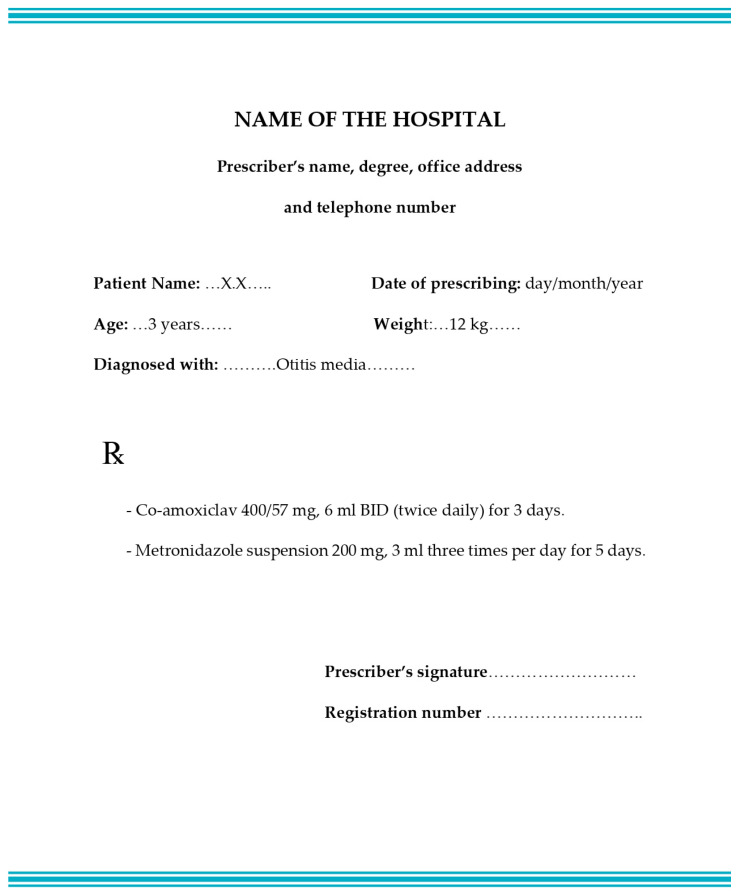
Shape of the standard prescription used in the simulated visits.

**Table 1 pharmacy-11-00006-t001:** Details of the study scenario.

DRP Under Study	Patient Data and Diagnosis	Prescription Details	The Problem	Comments
Wrong dose	Prescription for a 3 year old boy, weighing 12 kg, diagnosed with otitis media	Co-amoxiclav 400/57 mg 6 mL BID (twice daily)	Overdose	The accurate dose for this patient according to weight is 2.5 mL BID, maximum dose of 5 mL in severe infection
Wrong duration		Co-amoxiclav 400/57 mg for 3 days	The use is only for 3 days	The duration of this medication is 5 to 7 days
Wrong indication		Metronidazole suspension 200 mg 3 mL three times per day for 5 days	Prescription includes metronidazole	Improper indication of metronidazole as it does not treat otitis media

**Table 2 pharmacy-11-00006-t002:** Assessment form to assess the ability of community pharmacists (CPs) in identifying and resolving Drug-related problems (DRPs) in a pediatric prescription.

Identification of DRPs		
DRP Understudy	Coding	
	Yes	No
Did the CP identify the wrong dose?	1	0
Did the CP identify the wrong duration?	1	0
Did the CP identify the wrong indication?	1	0
**The response of the CP to DRPs**		
What was the response of the CP regarding the wrong dose?		
What was the response of the CP regarding the wrong duration?		
What was the response of the CP regarding the wrong indication?		
**Time of dispensing**		
The time (in second) spent by CP during the dispensing process		

**Table 3 pharmacy-11-00006-t003:** Socio-demographic characteristics of the community pharmacists (*n* = 235).

Characteristics of the Respondents	Frequency (%)
**Age groups**	
21–25 years	96 (40.9)
26–30 years	84 (35.7)
31–35 years	38 (16.2)
36–40 years	14 (6)
Over 40 years	3 (1.3)
**Gender**	
Male	72 (30.6)
Female	163 (69.4)
**Highest professional qualification**	
B. Pharm. degree	181 (77)
MSc. degree	54 (23)
**Number of years of experience as a community pharmacist**	
Less than 2 years	58 (24.7)
2–5 years	68 (28.9)
6–10 years	64 (27.2)
More than 10 years	45 (19.1)

**Table 4 pharmacy-11-00006-t004:** Identification of DRPs by community pharmacists.

Type of DRP	The Number (%) of CPs Identified the DRP	The Number (%) of CPs Who Did Not Identify the DRP
**Wrong dose**	10 (4%)	225 (96%)
**Wrong duration**	45 (19%)	190 (81%)
**Wrong indication**	6 (3%)	229 (97%)
**CPs who identified one, two or three DRP**		
**Type and number of DRP**	**Number of CPs**	**Percent**
**Wrong dose**	2	0.9%
**Wrong duration**	37	15.7%
**Wrong indication**	3	1.3%
**Wrong dose and duration**	6	2.6%
**Wrong duration and indication**	1	0.4%
**Wrong dose, duration and indication**	2	0.9%

**Table 5 pharmacy-11-00006-t005:** Actions taken by the community pharmacists who identified DRPs.

DRP	Pharmacists’ Action	Frequency (%)
**Wrong dose**	Dispensed as it is	0
	Corrected the dose	10 (100%)
	Referred to physician	0
**Wrong duration**	Dispensed as it is	0
	Corrected the duration	45 (100%)
	Referred to physician	0
**Wrong indication**	Dispensed as it is	0
	Refused to dispense	6 (100%)
	Referred to physician	0

## Data Availability

All data used and analyzed during the current study are available from the corresponding author on reasonable request.

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
