# Peer review of "Evaluation of Community Pharmacists’ Competences in Identifying and Resolve Drug-Related Problems in a Pediatric Prescription Using the Simulated Patient Method"

_pharmacy, 2022, doi:10.3390/pharmacy11010006_

Round 1

Reviewer 2 Report

Peer Review – 2025002

Evaluation of Community Pharmacists’ Competences in Identifying and Resolve Drug-Related Problems in a Pediatric Prescription Using the Simulated Patient Method

Was the Board of Pharmacy (or appropriate regulatory agency) aware of this study and it approved the use of the fake prescriptions?

Was the prescriber for each of these prescriptions an actual practitioner in the area?

What is the law in Sudan regarding what be changed on a prescription without contacting the prescriber?

What were the COVID-19 related restrictions in Khartoum at the time of the study? Could any COVID-19 related restrictions have influenced these results?

Internal Validity

Issues that need to be addressed:

Was a particular time period used during the day to present the prescription to the pharmacy?  If not, did you do a subgroup analysis to see if that may or may not have influenced the time the pharmacists spent with the prescription encounter?  In other words, was less time spent with the individual and the filling process during busy hours versus other time periods?

Did all of the pharmacies use a computer system as part of their filling process?  If so, what are the minimum standards in the Sudan for these computer systems, such as built in safe guards related to DRPs?

External Validity

While the information might be generalizable, you only have information about this specific geographic location. Therefore, any statement not specific to the location is not valid based on the data obtained during this study.  You addressed this issue in limitations area of the discussion, but your statements in the abstract and final conclusion are too general and should be specific to the study location. Right now it reads as if this is the standard of care throughout the world and the same solution is required to fix the problem.

In the discussion area, can you make any statements regarding the use of computer technology and how it might have influenced the results in the UAE and USA studies?  Were the prescriptions that were used in the studies from the UAE and USA have the same type of DRP?  If not, that needs to be made cleared in your discussion. Based on your statement they were both done using a pediatric SP, but it is unclear what type of DRP was being evaluated. Adding dates within the text might also help, since information from a study conducted in several years ago may no longer be valid because of changes in computer programming or its application within particular practice locations?

Other issues:

Page 6 line 216, why was the author this study listed when you did not include that information in the previous comments about other studies?

Page 6 line 242, the last two sentences really don’t add anything to the paper. Plus you need to keep your comments focused on the geographic region of the study.  Discussing what the content of the prescribing information required by another county is not applicable to the study population.  You should only include these statements if they are within the required labeling for the study country.

Round 2

Reviewer 2 Report

Authors changes and comments are acceptable.